# Frontotemporal Dementia P301L Mutation Potentiates but Is Not Sufficient to Cause the Formation of Cytotoxic Fibrils of Tau

**DOI:** 10.3390/ijms241914996

**Published:** 2023-10-08

**Authors:** Kuang-Wei Wang, Gary Zhang, Min-Hao Kuo

**Affiliations:** Department of Biochemistry and Molecular Biology, Michigan State University, East Lansing, MI 48824, USA; wangkua2@msu.edu (K.-W.W.); zhangga6@msu.edu (G.Z.)

**Keywords:** Alzheimer’s disease, tauopathy, P301L, hyperphosphorylated tau, apomorphine, frontotemporal dementia

## Abstract

The P301L mutation in tau protein is a prevalent pathogenic mutation associated with neurodegenerative frontotemporal dementia, FTD. The mechanism by which P301L triggers or facilitates neurodegeneration at the molecular level remains unclear. In this work, we examined the effect of the P301L mutation on the biochemical and biological characteristics of pathologically relevant hyperphosphorylated tau. Hyperphosphorylated P301L tau forms cytotoxic aggregates more efficiently than hyperphosphorylated wildtype tau or unphosphorylated P301L tau in vitro. Mechanistic studies establish that hyperphosphorylated P301L tau exacerbates endoplasmic reticulum (ER) stress-associated gene upregulation in a neuroblastoma cell line when compared to wildtype hyperphosphorylated tau treatment. Furthermore, the microtubule cytoskeleton is severely disrupted following hyperphosphorylated P301L tau treatment. A hyperphosphorylated tau aggregation inhibitor, apomorphine, also inhibits the harmful effects caused by P301L hyperphosphorylated tau. In short, the P301L single mutation within the core repeat domain of tau renders the underlying hyperphosphorylated tau more potent in eliciting ER stress and cytoskeleton damage. However, the P301L mutation alone, without hyperphosphorylation, is not sufficient to cause these phenotypes. Understanding the conditions and mechanisms whereby selective mutations aggravate the pathogenic activities of tau can provide pivotal clues on novel strategies for drug development for frontotemporal dementia and other related neurodegenerative tauopathies, including Alzheimer’s disease.

## 1. Introduction

Tauopathies are neurodegenerative diseases manifested by tau inclusions in neurons, glia, or astrocytes as their shared hallmark biomarker. The brains of patients with tauopathies, such as Alzheimer’s disease, frontotemporal lobar degeneration (FTLD), Pick’s disease, corticobasal degeneration, or progressive supranuclear palsy, show protein aggregates of the microtubule-associated protein tau (also known as MAPT) [1]. Although there are clear distinctions in the clinical manifestations of these disorders, tau is considered an important player in these diseases.

Tau is a microtubule-associated protein [2], encoded by the MAPT gene. Though its microtubule stabilizing activity is well-known in vitro, the normal, biological role of tau remains unclear [3]. Tau can possess a remarkable degree of posttranslational modification, including N- and O-glycosylation, ubiquitination, truncation, glycation, and oxidation, amounting to 63 unique posttranslational modifications that have been reported in wild-type mice [4]. Despite the diversity in posttranslational modifications, a cohort of neurodegenerative tauopathies share the trait of the deposition of hyperphosphorylated tau, whether wildtype or bearing a mutation [5,6], suggesting that abnormal phosphorylation of tau plays one of the key factors in the disease.

Tauopathies can be classified into primary or secondary tauopathies. In secondary tauopathies such as AD, tau dysregulation is thought to be an effect of other pathological proteins, most notably the aggregation of amyloid-β (Aβ) peptides in the senile plaques of AD patients [7]. While relatively small cohorts of familial early-onset AD are associated with increased synthesis and aggregation of Aβ caused by gene mutations or variants, most AD cases are sporadic and late-onset [8,9]. In both familial and sporadic cases [10], increases in both tau protein level and phosphorylation state have been observed in AD patients [11,12]. Hyperphosphorylated tau can assemble and propagate in neurons through a prion-like molecular process. The aggregation of tau protein can be induced by the intracerebral infusion of pathological tau into transgenic mice expressing human tau or, to a lesser extent, into wild-type, non-transgenic mice [13,14,15]. Furthermore, the spatiotemporal distribution of hyperphosphorylated tau in filaments or neurofibrillary tangles closely correlates with the progression of cognitive impairments in human patients [12,16,17], strongly suggesting a pathogenic role for abnormally phosphorylated tau. Although Aβ and tau both are relevant in AD-associated neurodegeneration, primary tauopathies such as certain frontotemporal dementias (FTD; also termed tau-positive frontotemporal lobar degeneration, FTLD-tau [18]), Pick’s disease, progressive supranuclear palsy, cortical basal degeneration, and argyrophilic grain diseases progress in the absence of overt Aβ pathology [7]. It is therefore increasingly accepted that dysregulated tau plays a direct role in neurodegeneration in primary and secondary tauopathies.

FTLD-tau is caused by mutations mapped to tau, thereby demonstrating that tau anomalies can drive neurodegeneration without Aβ. One of the most prevalent FTLD mutations is P301L (proline-to-leucine mutation) [19,20]. Mechanistically, certain mutations reduce microtubule binding and facilitate tau polymerization [21], while others affect the association of tau with the plasma membrane [22]. Both wildtype (WT) and FTLD mutant forms of tau have been expressed in mice. The transgenic expression of human P301L tau facilitates neurofibrillary tangle (NFT) formation at an early age, yet human WT tau-expressing mice showed delayed NFT formation [23]. These animal studies have been complemented by other investigations in cell lines that have shown, for example, that transient overexpression of WT or disease-linked mutations in tau (R406W, P301L, ΔN296) had a dramatic effect on the microtubule cytoskeleton [24]. Together, these reports demonstrated that mutations in tau can drive neurological pathology faster than wildtype tau in vivo. The P301L mutation has been widely reported for its effects on facilitating tau-mediated pathology and was used in the PS19 transgenic mouse model [25] of tauopathy. However, how it affects the aggregation of tau molecules and stimulates neuronal loss remains to be delineated.

Pathologically, AD is associated with the accumulation of misfolded proteins, metabolic derangements, elevated oxidative stress, and neuroinflammation in the brain [26]. Particularly, the progressive accumulation of hyperphosphorylated tau protein or Aβ peptides represents a major cytotoxic challenge in AD. In mammalian cells, accumulation of unfolded or misfolded proteins in the endoplasmic reticulum (ER) triggers an ER stress response or unfolded protein response (UPR), mediated by three ER-localized stress sensors: inositol-requiring enzyme 1α (IRE1α), protein kinase (PKR)-like endoplasmic reticulum kinase (PERK), and activating transcription factor 6 (ATF6), to help the cells survive from and adapt to ER stress conditions [27,28]. Among the ER stress sensors, PERK is activated to phosphorylate the eukaryotic translation initiation factor eIF2α and thus attenuates general protein synthesis to reduce the luminal client load in response to ER stress. However, under prolonged ER stress or acute injuries to the ER, PERK-mediated UPR induces proapoptotic transcription factors, including ATF3, ATF4, and CHOP (C/EBP homologous protein), as well as BH3-only BCL2 family members, such as NOXA and BIM [29]. CHOP can induce cell cycle arrest or apoptosis by regulating the expression of multiple genes encoding proapoptotic factors, including DR5 (death receptor 5), TRB3 (Tribbles homolog 3), and CAVI (carbonic anhydrase VI). CHOP also contributes to apoptosis through activating ERO1α, an ER oxidase that promotes hyperoxidization of the ER. Additionally, ER stress, UPR, and inflammatory responses are interconnected through various mechanisms, including the production of reactive oxygen species (ROS), the release of calcium from the ER, the activation of the transcription factor nuclear factor-κB (NF-κB), and the mitogen-activated protein kinase known as JNK (JUN N-terminal kinase) [30]. The ER stress response promotes the production of proinflammatory cytokines in macrophages, fibroblasts, astrocytes, or epithelial cells associated with autoimmune, metabolic, and neurodegenerative diseases [31].

We previously reported the synthesis and characterization of recombinant hyperphosphorylated tau by the PIMAX approach (Protein Interaction Modules-Assisted Function X) [32]. PIMAX is a versatile method that enables the synthesis of recombinant proteins in *E. coli* with a desired post-translational modification. Hyperphosphorylated tau produced by PIMAX bears GSK-3β phosphorylation marks that recapitulate many of those found in AD, which render the underlying tau prone to aggregate into a cytotoxic species without any artificial inducer such as heparin or arachidonic acid [33]. In this study, we produced hyperphosphorylated P301L tau (P301L p-tau), hyperphosphorylated wildtype tau (WT p-tau), and unphosphorylated P301L tau (P301L tau) with the PIMAX approach to compare their characteristics, which includes assessing cytotoxicity and aggregation ability. We also show their differences in the activation of ER stress, UPR, and the inflammatory response in neuroblastoma cells. In addition, damages in the cytoskeleton known to activate these stress responses are compared between the treatments of P301L p-tau, WT p-tau, and P301L tau. These experimental results reveal the relevance that studying mutant forms of tau has to drug screening and therapeutic efforts.

## 2. Results

### 2.1. Production of Hyperphosphorylated Tau Using the PIMAX Approach

One of the challenges of studying tau pathology is the prominent post-translational modifications, most notably hyperphosphorylation [34], that are absent in most of the recombinant tau protein samples used in a variety of mechanistic studies. Contrary to common recombinant protein production approaches, PIMAX utilizes a pair of heterodimerizing domains to facilitate the interaction between two proteins of interest. If the two proteins of interest consist of a protein-modifying enzyme and its substrate, proteins with a desired post-translational modification can be easily produced [32]. Figure 1A shows how PIMAX was used to produce hyperphosphorylated tau. In short, the leucine zipper domain of Fos and Jun proteins was fused to GSK-3β and 1N4R tau (Figure 1A). Bacterial co-expression of these two chimeric proteins resulted in heterodimerization of Fos and Jun leucine zipper domains, thereby facilitating the interaction between tau and GSK-3β and hence the highly efficient phosphorylation of tau. The P301L mutation is in the second microtubule binding domain (R2) and only affects the 4R (4 repeat) isoform since the relevant exon is spliced out of the 3R isoform [35] (Figure 1B). The P301L mutation was introduced into the 1N4R tau PIMAX plasmid, and the resultant mutant tau was expressed and purified along with the wildtype p-tau. SDS-PAGE with Coomassie blue staining revealed a phosphorylation-dependent mobility shift of both wildtype and P301L tau when compared to the ones without the GSK-3β kinase (Figure 1C). As shown here and in numerous prior reports, the mobility of tau in SDS-PAGE was substantially different from the calculated size. The full-length 1N4R is calculated to be 42.67 kDa, whereas the estimated unphosphorylated tau mobility on a denaturing protein gel is near 60 kDa. Hyperphosphorylation further retarded the mobility of both wildtype and P301L p-tau to about 65 kDa. Abnormal phosphorylation in tau is thought to initiate different mechanistic and toxic characteristics of the underlying tau protein [1]. This abnormal phosphorylation can take form in the pattern of Ser/Thr phosphorylation sites, which suggests that phosphorylation of certain residues may trigger toxicity [36]. To see whether the P301L mutation caused a significant deviation in tau phosphorylation, we used mass spectrometry to map the phosphorylation sites of the WT and P301L hyperphosphorylated tau. The MS results showed that all but one (S262) phosphorylation site mapped to the P301L p-tau was also present in the wildtype p-tau (Figure 1D). There are 14 additional residues scattered throughout the domains that were phosphorylated in the WT but not the P301L p-tau, suggesting structural differentiation of these two tau species extended beyond the region surrounding residue 301. Phosphorylation of tau is the most prevalent post-translational modification of tau inclusions in many tauopathies. Among the P301L p-tau phosphorylation sites, sixteen out of twenty-three (T153, T175, T181, S191, S202, S214, T217, T231, S235, S262, S289, S356, S396, S400, T403, and S404) are associated with AD-paired helical filaments (PHFs) [37], verifying the disease relevance. However, the detailed phosphorylation mapping data of tau-bearing P301L mutations in patients have not been reported.

### 2.2. P301L Mutation Enhances the Propensity to Aggregate and to Damage Cells by Hyperphosphorylated Tau

A hallmark character of pathological tau is its ability to aggregate and cause cell death [33]. Tau, which normally functions to stabilize axonal microtubules, goes awry after elevated phosphorylation and assumes the ability to form fibrillar aggregates [38]. Although which form of tau aggregates is the cause of cytotoxicity is under debate [39], the presence of these characteristics can be useful for comparing the activity of different forms of tau, for tauopathies are manifested by certain selective isoforms of hyperphosphorylated tau that appear to differentially affect neurons, astrocytes, and glial cells [40]. To examine whether the P301L mutation confers specific molecular characteristics that may be different from the wildtype p-tau, we first tested the inducer-free aggregation of mutant P301L p-tau along with the wildtype and the unphosphorylated counterparts with thioflavin S (ThS), which emits fluorescence upon binding the amyloid structure that is shared by many proteopathy proteins, including tau, to assess whether the P301L mutation could affect the kinetics of tau aggregation [41]. At 6 µM, P301L p-tau consistently showed a higher magnitude of ThS fluorescent signal than the WT p-tau and P301L tau (Figure 2A). This dose of p-tau was selected because there is ~10 μM total tau in the frontal cortex of terminal AD [11], of which 40–70% was estimated to be in a non-fibril oligomeric state [42]. It was also close to the pathological state but lower than most published aggregation assays using unphosphorylated tau. P301L p-tau exhibited the largest net increase of 130 ThS fluorescence units, while WT p-tau and P301L tau increased by 90 and 30 units, respectively (Figure 2B). All four tau species reached their maximal ThS intensity in about 30 min. The ThS fluorescence was statistically significant for P301L p-tau compared to P301L tau (without phosphorylation).

We next examined whether the P301L mutation would affect the ability of tau to cause cell death. Neuroblastoma (SH-SY5Y) cells, a model widely used for studying neurodegenerative disease, were treated with 2 µM of P301L p-tau, p-tau, and P301L tau. The LD_50_ of p-tau sometimes varies slightly, depending on the batch of protein or the passage number of cells. Two µM of p-tau gave consistent results in our cell-killing experiments. After 20 h of incubation, both P301L and WT p-tau caused a striking reduction in relative cell viability, whereas the unphosphorylated P301L tau was benign under the same test conditions (Figure 2C). Noticeably, the P301L mutant p-tau exhibited more severe cytotoxicity than the WT p-tau, consistent with many animal studies in which transgenic overexpression of this and similar mutant alleles of tau resulted in accelerated tau pathology and cognitive impairments [25].

### 2.3. P301L p-Tau Has Enhanced Seeding Activity That Induces Tau Aggregation in Primary Neurons

The seeding activity of tau is a well-known feature of pathological tau [43]. We suspected that this mutation conferred stronger seeding activity on the underlying p-tau to incorporate unphosphorylated tau into fibrils. To test this hypothesis, we treated mouse primary neurons with different p-tau seeds along with an excessive amount of unphosphorylated tau. We have previously reported that hyperphosphorylated tau produced by the PIMAX system can nucleate tau aggregation and exacerbate tau cytotoxicity in SH-SY5Y cells [33]. To better model the neuropathology associated with tau, we used mouse primary neurons for this set of experiments. Primary neurons were treated at DIV (days in vitro) 9 with a mixture of 0.1 µM p-tau, wildtype or P301L, and 0.9 µM of unphosphorylated, wildtype tau for three days, followed by ThS staining for fibrils in cells. Neurons treated with P301L p-tau (0.1 µM) and WT tau (0.9 µM) showed the strongest ThS signals, whereas the wildtype p-tau seeds with tau or the unphosphorylated tau (1.0 µM) alone showed only weak fibril staining (Figure 3). P301L p-tau seeds also seemed to induce abnormal neuron morphology with shrinkage in the cell body and fewer dendrites (arrow, Figure 3). These results confirmed that the P301L mutation promoted the ability of hyperphosphorylated tau to recruit benign tau molecules into a species that may be harmful to cells.

### 2.4. P301L p-Tau Forms Aggregate Structures More Rapidly Than WT p-Tau and P301L Tau

Mature paired helical filaments and straight filaments that make up neurofibrillary tangles are commonly thought to be of more pathological importance than amorphous aggregate structures [44]. However, studies suggest that the formation of neurofibrillary tangles is uncoupled to neuronal dysfunction or death [45,46]. To investigate how the P301L mutation might control tau aggregation, P301L p-tau, WT p-tau, and P301L tau were incubated without an artificial inducer to allow for the autonomous formation of aggregate structures. The morphology of these aggregates (Figure 4A) and the identity of tau were visualized by immunogold transmission electron microscopy (Figure 4B). As expected, there were no canonical neurofibrillary or paired helical structures observed during the short 3-day course of reactions (Figure 4A). However, all four tau species formed clear conglomerated entities that could be best described as fractals [33]. The overall sizes of these aggregates appeared to echo the ThS aggregation assay results shown in Figure 2, that is, the P301L p-tau exhibited the most efficient aggregation, followed by the wildtype p-tau and the two unphosphorylated tau. Between the two hyperphosphorylated proteins, despite the fact that the P301L mutant was phosphorylated at a smaller set of residues, this mutant p-tau showed aggregates even without any incubation (0 h incubation, Figure 4A), suggesting that the P301L mutation had caused the underlying hyperphosphorylated tau to form larger assemblies even before the preparation process was completed.

### 2.5. P301L p-Tau Adopts a Specific Conformation Sensitive to Bacterial Endopeptidases

Upon inspecting different preps of the wildtype and the P301L p-tau, we frequently noticed a ~40-kDa band associated specifically with the mutant form of p-tau (Figure 5A). For conformational alterations of tau following its abnormal phosphorylation is generally regarded as an underpinning for the formation of neurotoxic aggregates that wreak havoc in the tauopathies, and that the P301L p-tau consistently showed stronger propensities of aggregation and killing tissue culture cells (Figure 2, Figure 3, and sections below), we wondered whether the presence of a smaller species of the P301L p-tau was indicative of a unique conformation of this mutant form that was particularly amenable to proteolysis. To answer this question, we set out to first verify the identity of this 40-kDa band by immunoblotting analysis with the DA9 antibody [47], which recognized tau independently of its phosphorylation status. Figure 5B shows that this antibody detected both the full-length (~65 kDa) and this 40-kDa species, but not a ~37-kDa band seen in both wildtype and P301L p-tau preps. This result confirmed that a 40-kDa species of tau was present specifically in the P301L p-tau preparation. We then used mass spectrometry to investigate how this smaller form of p-tau might be generated. Figure 5C shows that this band contained two discontinuous portions of p-tau, residues 6–67 and 137–412 of the 1N4R isoform used in this study (corresponding to residues 6–67 and 166–441 of 2N4R tau). The missing internal portion spanned the last 8 residues of the N1 domain (aa 44–74) and the first 14 amino acids of the proline-rich domain (aa 123–213). The calculated molecular weights of these two peptides are 6.79 kDa and 29.45 kDa. Given the significant deviation of tau mobility from the calculated molecular weight, it is difficult to ascertain at the moment whether the two peptides were resolved as a tightly associated complex or that the smaller peptide (aa 6–67) was simply a contaminating species. Either way, it was very likely that a bacterial endopeptidase was responsible for the cleavage of P301L p-tau during the preparation.

If the hypothesis of P301L-specific proteolytic digestion by a bacterial endopeptidase was true, one could then further extrapolate and posit that this mutation caused a conformational alteration that consequently exposed such a cleavage site for the observed proteolysis. To explore this scenario further, we used a tau conformation antibody, MC1 [48]. MC1 reactivity is considered one of the earliest pathological alterations of tau in AD [49]. Figure 5D shows the Western dot-blot analysis in which four different tau species were serially diluted and pipetted to a piece of nitrocellulose membrane without pretreating with the denaturing detergent SDS or any reducing reagent such as β-mercaptoethanol or dithiothreitol to preserve the native conformation of these proteins. The general tau antibody DA9 showed that comparable amounts of mutant and wildtype p-tau and tau preps were loaded into the membrane. However, the MC1 reactivity was apparently stronger in the P301L p-tau than in the WT p-tau and P301L tau, followed by WT tau, suggesting that a conformational change had taken place in the mutant proteins. The increased MC1 epitope, and hence a disease-related conformation, may be an underlying reason for the enhanced cytotoxicity of p-tau (Figure 2 and below).

### 2.6. P301L p-Tau Exhibits the Strongest Induction of ER Stress-Associated Pro-Apoptosis

In eukaryotic cells, a number of biochemical and pathophysiological challenges or the presence of aberrant proteins can interrupt the protein folding process in the endoplasmic reticulum (ER), the organelle where protein folding and assembly take place, and therefore cause the accumulation of unfolded or misfolded proteins, a condition called “ER stress”, that leads to the activation of the ER stress response or unfolded protein response (UPR) to restore protein homeostasis [27,50].

To understand the mechanisms by which P301L p-tau inflicts cellular damage and death, we treated the SH-SY5Y cells with a sub-lethal dose (0.5 µM) of p-tau, P301L p-tau, or P301L tau for 24 h. Quantitative real-time PCR (qPCR) analyses with the total RNAs isolated from the treated cells were performed to determine the expression profile of the genes encoding ER stress markers and major components of the ER stress-associated pro-apoptosis pathway. In response to the p-tau or P301L p-tau challenge, expression levels of the ER chaperone BiP/GRP78, a master regulator of the ER stress response [28], in the neuroblastoma cells were significantly increased compared to those of the cells treated with vehicle (Figure 6A), demonstrating that p-tau or P301L p-tau is a potent trigger for ER stress. In contrast, induction of BiP/GRP78 in SH-SY5Y cells by P301L tau treatment was marginal, if any, compared to that in the cells treated with vehicle (Figure 6A–E). Furthermore, we examined the levels of the transcripts encoding the mediators or markers of ER stress-associated pro-apoptotic pathways, including ATF4, CHOP, TRB3, and GADD34, expressed by SH-SY5Y cells in response to tau treatments. While p-tau treatment induced expression of ATF4, CHOP, TRB3, and GADD34 in SH-SY5Y cells, P301L p-tau challenge led to expression of ATF4, CHOP, TRB3, and GADD34 to much higher levels (Figure 6B–E). In contrast, P301L tau without phosphorylation can only slightly induce expression of these ER stress-associated pro-apoptotic factors. These results are consistent with the CCK8 cell viability assay results shown in Figure 2C and suggest that P301L p-tau’s stronger cytotoxicity was due at least partly to the more potent ER stress-inducing activity.

### 2.7. Apomorphine Antagonizes Both the Aggregation and Cytotoxicity of P301L p-Tau

To see whether the highly pathogenic P301L p-tau can be controlled pharmacologically, we examined the potential of apomorphine to counter P301L p-tau aggregation and cytotoxicity in vitro. Apomorphine, a dopamine agonist for the treatment of Parkinson’s disease symptoms [51], was found in our Alzheimer’s disease drug discovery pilot screen to be a potent small-molecule compound with significant power in ameliorating both the aggregation and cytotoxicity of the wildtype p-tau [52]. Considering the enhanced detrimental impact of hyperphosphorylated tau due to the P301L mutation, this protein could serve as a potential target for the advancement of therapeutic strategies for tauopathies including Alzheimer’s disease (AD), frontotemporal lobar degeneration with tau aggregates (FTLD-tau), and various other tau-related disorders. To this end, we first examined whether apomorphine could effectively influence the formation of the amyloid structure of P301L p-tau in our standard aggregation assay. Indeed, apomorphine neutralized P301L p-tau aggregation to a similar degree as WT p-tau, showing that while the P301L mutation makes p-tau more aggressive, it did not change mechanistically the way of fibril formation, and that apomorphine impacted both forms of p-tau similarly (Figure 7A).

To further assess whether apomorphine diminished P301L p-tau toxicity in ER stress elicitation, SH-SY5Y cells were treated with 0.5 µM P301L p-tau in the presence or absence of 5 µM apomorphine for 20 h, followed by qPCR analysis. P301L p-tau-induced expression of the transcripts encoding the ER chaperone BiP/GRP78 as well as the ER stress-associated pro-apoptotic factors ATF4, CHOP, TRB3, and GADD34 was significantly lessened by apomorphine (Figure 7B–F). This result indicates that P301L p-tau-caused ER stress-associated apoptosis of cells could be antagonized by apomorphine.

### 2.8. P301L p-Tau, but Not P301L Tau, Can Disrupt the Cellular Cytoskeleton

The microtubule cytoskeleton is a dynamic structure in cells that is intimately linked, functionally and physically, to many cellular functions and organelles, including ER [50]. Disruption of the cytoskeleton may therefore cause major damage to cells. Given the conspicuous effects of p-tau on ER functions, we suspected that ER impairments in P301L p-tau-treated cells at least partly resulted from cytoskeleton dysfunction. To test this notion, we used a fluorescence probe for phalloidin to examine the cytoskeleton structures of cells treated with 0.5 µM P301L p-tau for 24 h. Indeed, P301L p-tau treatment caused drastic disorganization of the cytoskeleton, and the F-actin fluorescence signals were nearly completely lost (Figure 8A). In contrast, the unphosphorylated P301L tau failed to impact the cytoskeleton morphology appreciably (Figure 8B), consistent with its lack of cytotoxicity as shown in Figure 2 and Figure 4 above. In the presence of apomorphine, cytoskeleton disintegration was completely suppressed (Figure 8A, green bar). Importantly, the prevention of P301L p-tau-induced cytoskeleton disorganization, in conjunction with the suppression of ER stress and UPR, by apomorphine resulted in full protection of cell viability (Figure 8C). Together, these results demonstrated the potent cytoprotective activity of this compound in combating P301L p-tau.

## 3. Discussion

Deposition of hyperphosphorylated tau in the brain is a telling pathological trait shared by neurodegenerative tauopathies, including primary tauopathies such as tau-positive frontotemporal lobar dementia and Pick’s disease and secondary tauopathies such as Alzheimer’s disease and chronic traumatic encephalopathy [7]. The spatiotemporal distribution of tau precipitates correlates with the clinical manifestation of cognitive and neurological deficits in AD [53] and other disorders [54], suggesting a causal role for abnormal tau, a notion that is supported by the identification of P301L and other mutations in the MAPT gene in frontotemporal dementia with Parkinsonism linked to chromosome 17 (FTDP-17) [55]. On the other hand, despite the frequent exploitation of some of these FTDP-17 tau mutations in both in vitro and in vivo models [56], little is known as to whether and how these mutations confer characteristics that deviate from the wildtype tau protein. Furthermore, few studies have been undertaken specifically on hyperphosphorylated P301L mutant tau up to this point. Here we show that, indeed, the FTLD-related P301L mutation potentiates the hyperphosphorylated tau to aggregate and form cytotoxic species that instigate different cellular stress pathways and cause cell death. The unphosphorylated P301L tau does not appear to be harmful to cultured cells. It is therefore surmised that the combination of hyperphosphorylation and the P301L mutation (or likely a similar disease-relevant sequence alteration) brings in a very detrimental species of tau underlying progressive neurological pathology seen in humans and model animals.

P301L and P301S are among the most prevalent FTLD-associated mutations and have been used in multiple animal models [57]. However, how P301L or other FTLD mutants, such as R406W and ΔK280, cause tau to trigger or facilitate neurodegeneration is still unclear. One of the difficulties in the field that contributes to the knowledge gap about P301L pathogenesis is the lack of pathologically relevant species for molecular and mechanistic studies. Tau protein from post-mortem samples is frequently tainted with other biomolecules and presents the complexity of different PTM [32]. Recombinant tau, wildtype or mutant, typically lacks the disease-relevant hyperphosphorylation mark. This work is the first to examine the molecular behaviors of hyperphosphorylated tau bearing the P301L mutation. P-tau produced by PIMAX exhibits characteristics relevant to the pathology of neurodegeneration, which include multiple phosphorylation sites associated with the disease [33,37], inducer-free fibril formation, and clear cytotoxicity at low-micromolar concentrations. Taking advantage of this system, a hyperphosphorylated P301L tau was produced and characterized herein. This work reveals several key differences between the wildtype and the mutant p-tau that shed light on the possible mechanism underlying the FTLD neuropathology instigated by the mutant tau and provides a platform for future comparative studies of different isoforms (e.g., 3R vs. 4R), wildtype or mutant, of hyperphosphorylated tau in tauopathies.

The mass spectrometry phospho-mapping results (Figure 1D) show that, while the wildtype tau possesses a broader set of phosphorylation targets, the P301L mutant p-tau exhibits molecular and cytotoxic characteristics comparable to or even more effective than its wildtype counterpart. We thus suggest that the P301L hyperphosphorylated tau presents a core of phosphorylation that underlies the unique molecular characteristics of pathological tau, including autonomous aggregation, potent seeding activities, triggering ER stress, UPR, cytoskeletal disintegration, and cell death. Additionally, the recombinant P301L p-tau shows a 40-kDa proteolytic product immediately after its purification from a bacterial lysate. Considering that the reactivity with the MC1 tau conformation antibody was stronger in the mutant form of both p-tau and tau, we favor the hypothesis that the proline-to-leucine mutation at position 301 changes the three-dimensional structure of tau, thereby presenting a target site for a bacterial protease and enhancing the MC1 recognition. In addition, the P301L mutation-induced structural alteration may also restrict the accessibility of residues that are otherwise phosphorylated under the same PIMAX expression condition by GSK-3β. It is, however, worth noting that whereas the MC1 epitope is an early marker of tau pathology in AD, as well as in FTLD patients [58] and model mice [59], this association by no means demonstrates a pathologically causal role of the MC1-positive conformation. Single-molecule experiments demonstrated that liquid-liquid phase separation of tau to form aggregates in vitro is preceded by relaxation of the N’ and C’ regions, therefore exposing the core repeat domains for inter-molecular interactions and fibril formation, and that, importantly, the P301L mutation expedites this process [60]. These biophysical experiments provide molecular support for some of the observations presented in this work. Due to the uniquely rigid structure of peptidyl proline residues, it is not surprising that P301L impacts the spatial organization of tau. However, there are as many as 42 proline residues in the 2N4R tau, and Pro301 is not one of the seven PS or PT sites that may be regulated by Pin1-controlled cis-trans configuration [61]. It therefore remains an open question as to how hyperphosphorylation and the P301L mutation together contribute to the cytotoxic behaviors that are likely the underpinning of the neurodegeneration of frontotemporal lobar degeneration.

Besides the potential conformational alteration discussed above, Ser262 phosphorylation is yet another peculiar characteristic of P301L p-tau. S262 is uniquely phosphorylated in the P301L mutant within the PIMAX context (Figure 1). From multiple independent expression and mass spectrometry mapping of different wildtype p-tau isoforms produced by the PIMAX approach (1N4R, 2N4R, 0N3R, and 0N4R), Ser262 was rarely found to be phosphorylated by GSK-3β when a wildtype tau was used (Hagar et al., manuscript in preparation). GSK-3 kinases, including GSK-3α and -3β, favor a consensus sequence of S/TXXXS/T^P^, where the first S/T(underlined) is the site of phosphorylation that can be stimulated by a priming phosphorylation event at the +4 position (superscript-P) [62]. GSK-3 belongs to the proline-directed CMCG family kinases [63] that favor Ser/Thr residues next to a proline. A substantial portion of the phosphorylation sites shown in Figure 1D fit either or both GSK-3β preferred sites. However, Ser262 (KIG^262^STENLK) belongs to neither, suggesting the possibility that the P301L mutation, in addition to exposing a protease site that generates the 40-kDa species, positions Ser262 in a way that makes it phosphorylatable by GSK-3β. As to the function of S262 phosphorylation, it was reported that a S262D phospho-mimicking mutation promoted GSK-3β-mediated tau phosphorylation at other sites [64]. However, aspartic acid substitution creates a constitutive, negatively charged sidechain. A carboxylic acid also differs in three-dimensional space from a phosphate. The results of Ser-to-Asp or Thr-to-Glu experiments therefore need to be interpreted with great caution. Intriguing as it is, whether the molecular differentiation between the P301L and the wildtype p-tau can be solely ascribed to Ser262 phosphorylation is an open question that will be interrogated comprehensively.

Thioflavin S-based aggregation reactions and transmission electron microscopy (Figure 2 and Figure 4) support that P301L hyperphosphorylated tau aggregates faster than the wildtype hyperphosphorylated tau does. However, canonical structures related to neurofibrillary tangles, such as paired helical or straight filaments, were not observed in either tau species after incubation for up to 3 days (Figure 4). Although one might anticipate that longer incubation would eventually lead to the formation of such structural entities, this work focuses exclusively on the early molecular behaviors of the hyperphosphorylated tau species, including the conspicuous cytotoxic responses detectable within 20 h of incubation. We believe this is more relevant clinically, for the ultimate goal of tauopathy drug development should rest on efficacious management of cognitive decline resulting from dysfunction or death of brain cells, not on reducing the burden of either Aβ or tau deposits. It is therefore exciting that the P301L hyperphosphorylated tau, albeit more damning than its wildtype counterpart, is amenable to the anti-aggregation and cytoprotective actions of apomorphine. Apomorphine is currently used to treat the off-episodes of Parkinson’s disease, thanks to its dopamine receptor agonist activity [51]. This compound was shown to preserve the memory of 3xTg AD transgenic mice that expressed human tau, a mutant PSEN, and a mutant APP [65]. Maintenance of neuronal insulin resistance in the brain was postulated to underlie the observed cognitive improvement in experimental mice [65]. However, it is also possible that the pathogenic tau expressed in this mouse model was antagonized by apomorphine, resulting in attenuation of neuropathology and, consequently, memory preservation. The effectiveness of apomorphine on both wildtype and mutant hyperphosphorylated tau indicates that future drugs that are able to prevent tau pathology in AD may also be valuable therapeutics for other tauopathies. Conversely, one may also envision that tauopathies such as traumatic brain injury and chronic traumatic encephalopathy that have a definitive time of onset may provide a useful platform for clinical trials of reagents for their prevention of dementia after the cerebral impact, and that such trial results may inform the usefulness of such reagents in treating AD.

## 4. Materials and Methods

Plasmid and expression protocol for 1N4R p-tau refer to the previous publication: https://doi.org/10.1074/mcp.O114.044412. Recombinant Protein Expression and Purification refers to the previous publication: https://doi.org/10.1007/s12035-020-02034-w. 150-mesh copper square grids were purchased from VWR. The DA9 and MC1 antibodies were generous gifts from Dr. Peter Davies. Goat-anti-Mouse IgG+IgM (10 nm gold) was a generous gift from Dr. Alicia Withrow. Cell Counting Kit-8 (CCK-8) was purchased from GlpBio. Fluorescein diacetate (FDA), propidium iodide (PI), and Thioflavin S were purchased from Sigma-Aldrich (Burlington, MA, USA). Neurobasal medium, B27 and N2 supplements, Penicillin-Streptomycin-Glutamine and the filamentous actin (F-actin) probe, Alexa Fluor 488 phalloidin (A12379), was purchased from ThemoFisher Scientific (Waltham, MA, USA).

### 4.1. Mass Spectrometry

The protein band of interest was excised from the SDS-PAGE gel and washed with 25 mM ammonium bicarbonate, followed by acetonitrile. Then, the sample was reduced with 10 mM dithiothreitol at 60 °C, followed by alkylation with 50 mM iodoacetamide at RT. Then, the protein was digested with sequencing-grade trypsin (Promega, Madison, WI, USA) at 37 °C for 4h and then quenched with formic acid, and the supernatant was analyzed directly without further processing. Half of each digested sample was analyzed by nano LC-MS/MS with a Waters M-Class HPLC system interfaced to a ThermoFisher Fusion Lumos mass spectrometer. Peptides were loaded on a trapping column and eluted over a 75 μm analytical column at 350 nL/min; both columns were packed with XSelect CSH 5μm C18 resin (Waters). The mass spectrometer was operated in data-dependent mode, with the Orbitrap operating at 60,000 FWHM and 15,000 FWHM for MS and MS/MS, respectively. The instrument was run with a 3s cycle for MS and MS/MS. Data were analyzed using Mascot software.

### 4.2. Coomassie Blue Staining

The SDS-PAGE gel was stained with 0.05% Coomassie blue R-250 (CBR) in 10% acetic acid and 25% isopropanol for 30 min at RT. The gel was then sequentially de-stained with 0.005% CBR in 10% acetic acid and 10% isopropanol, 0.002% CBR in 10% acetic acid, and finally 10% acetic acid for 30 min (with shaking).

### 4.3. Aggregation Assay

A solution of 6 µM tau protein with or without indicated compounds was mixed in 20 mM Tris pH 7.4, 1 mM DTT, and 20 µM ThS to a 10 µL final volume and placed into a 384-well black plate. Each plate was covered with transparent adhesive film and placed into a preheated 37-degree plate reader. ThS fluorescence was acquired every 10 min at 440/490 nm for 16 h.

### 4.4. Cytotoxicity Assay

For the cytotoxicity assay, 2000 SH-SY5Y cells were seeded on a 96-well plate in 100 µL DMEM/F12 media. 10 µL of protein solution with or without compounds was added to each well and placed back into the incubator for 20 h. Relative cell viability was determined using either cell counting kit 8 (CCK8) or FDA/PI staining and calculated by FDA positive cells/(FDA positive cells + PI positive cells).

### 4.5. Primary Neurons Isolation

E16 pregnant mice (C57BL/6) were ordered from Charles River Laboratories. The cerebral cortex was dissected from embryos and then passed through a 70 μm nylon mesh filter. Primary neurons were seeded on a 96-well plate at 5 × 10^5^ cells/mL density in 100 µL DMEM for 1 h at 37 °C with 5% CO_2_ for attachment. The medium was changed to Neurobasal medium (10% FBS, 1× N2 and B27 supplements, and 1× Penicillin-Streptomycin-Glutamine were added); 25% of the medium was changed with fresh Neurobasal medium every three days.

### 4.6. Thioflavin S Staining

A 100 µM final concentration of thioflavin S was added and stained the primary neurons for 10 min at 37 °C. The cell medium was changed to fresh Neurobasal medium. Neurons were imaged by a microscope with a 20× objective.

### 4.7. Transmission Electron Microscopy

First, 10 µM p-tau was incubated with or without compounds in 20 mM Tris buffer (pH 7.4) at 37 °C for an indicated time. The sample was 10-fold diluted and incubated with 2.5% glutaraldehyde for 5 min. Then, 20 µL of the sample was fixed on a 150-mesh copper square grid and negatively stained by 1% uranyl acetate for 10 s. The sample was imaged by JOEL 1400 Flash TEM. For immunogold staining, 20 µL of the sample was fixed on the grid with 2.5% paraformaldehyde for 5 min and blocked with 1% BSA at room temperature for 1 h. Then, the grid was incubated with primary antibody (DA9 antibody 1:1000) at 4 degrees overnight, followed by secondary 10 nm gold antibody (1:50) at room temperature for 2 h. The grid was then incubated with 2.5% glutaraldehyde for 15 min and stained with 1% uranyl acetate for 10 s. Then, the sample was imaged by JOEL 1400 Flash TEM.

### 4.8. Western Blotting

SDS-PAGE was transferred to the PVDF membrane. The membrane was blocked in 5% nonfat milk in TBST (20 mM Tris, pH 7.4, 150 mM NaCl, 0.1% Tween 20) for 1 h, and then incubated with the DA9 (at 1:10,000 dilution) for overnight at 4 °C. After washing three times in TBST, the membrane was incubated with HRP goat anti-mouse secondary antibody (at 1:10,000 dilution) for 1 h at room temperature. The membrane was washed again 3 times in TBST and developed with the Lumi-Light Western blotting substrate for 5 min.

### 4.9. Dot-Blot Assay

Two microliters of the samples were gently applied to a Nitrocellulose membrane. The membrane was blocked in 5% nonfat milk in TBST (20 mM Tris, pH 7.4, 150 mM NaCl, 0.1% Tween 20) for 1 h, and then incubated with the MC-1 monoclonal antibody (at 1:1000 dilution) or DA9 (at 1:10,000 dilution) for overnight at 4 °C. After washing three times in TBST, the membrane was incubated with HRP goat anti-mouse secondary antibody (at 1:10,000 dilution) for 1 h at room temperature. The membrane was washed again 3 times in TBST and developed with the Lumi-Light Western blotting substrate for 5 min.

### 4.10. qPCR

RNA was extracted from SH-SY5Y cells with TRIzol reagent (Thermo Fisher Scientific), and 500 ng of total RNA was reverse transcribed into cDNA with the High-Capacity cDNA Reverse Transcription Kit (Applied Biosystems, Foster City, CA, USA). mRNA levels were measured using RT-qPCR analysis using the SYBR green PCR Master Mix (Applied Biosystems). PCR was carried out using Applied Biosystems 7500 Real-Time PCR Systems. GAPDH, a housekeeping gene, was used as an internal control.

The primer sequences used in this study:

Hu BiP-F: CCTGGGTGGCGGAACCTTCGATGTG

Hu BiP-R: CTGGACGGGCTTCATAGTAGACCGG

Hu CHOP-F: GCCTTTCTCCTTTGGGACACTGTCCAGC

Hu CHOP-R: CTCGGCGAGTCGCCTCTACTTCCC

hu ATF4-F: TCAAACCTCATGGGTTCTCCA

hu ATF4-R: CACAGCCAGCCATTCGG

Hu TRB3-F: TACCTGCAAGGTGTACCCC

Hu TRB3-R: GGTCCGAGTGAAAAAGGCGTA

Hu GADD34-F: ATGATGGCATGTATGGTGAGC

Hu GADD34-R: AACCTTGCAGTGTCCTTATCAG

hu GAPDH-F: TGAAGGTCGGAGTCAACGG

hu GAPDH-R: AGAGTTAAAAGCAGCCCTGGTG

### 4.11. Cytoskeleton Staining

For cytoskeleton staining, cells were incubated with Alexa Fluor 488 phalloidin (Thermo Fisher), a high-affinity filamentous actin (F-actin) probe conjugated to green-fluorescent Alexa Fluor 488 dye (Thermo Fisher) for 1 h. The stained cells were mounted with DAPI-containing ProLong Antifade Mountant (Thermo Fisher) and observed using a Cytation Confocal Imaging Reader (Biotek/Agilent). To quantitatively analyze fluorescent intensities, cells of interest and regions of interest (ROI) were selected and measured for mean fluorescence intensity using ImageJ software. Corrected Total Cell Fluorescence (CTCF) was calculated based on: CTCF = Integrated Density − (Area of Selected Cell × Mean Fluorescence of Background readings).

### 4.12. Statistical Methods

Experiments were carried out with a minimum sample replicate size of *n* = 3. For comparisons between more than 2 groups (such as between WT tau, WT p-tau, P301L tau, and P301L p-tau), a one-way ANOVA was applied with Prism 9 (sample size *n* = 3). For comparisons between two groups, the Student’s t-test was applied for statistical analysis. Non-significant values are not marked to limit clutter in the data graphs. For the QPCR analysis and cytoskeleton staining quantification, biological replicates are displayed. For other experiments and figures, representative figures with technical replicates are displayed.

## Figures and Tables

**Figure 1 ijms-24-14996-f001:**
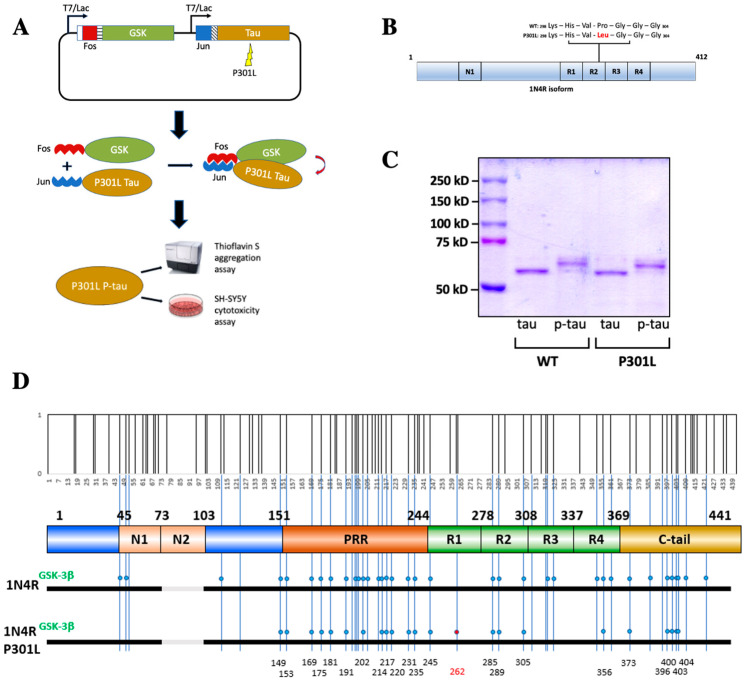
Production of hyperphosphorylated tau (p-tau). (**A**) PIMAX plasmid was designed with Fos-GSK and Jun-P301L 1N4R tau and expressed in *E. coli*. Leucine zipper proteins Fos and Jun modulate the protein interaction between GSK and P301L tau to efficiently produce p-tau. Purified tau and p-tau were assayed with Thioflavin S for aggregation kinetics and administered to SH-SY5Y cells to assay cytotoxic effects. (**B**) The P301L mutation (at the R2 domain) is known to cause frontotemporal dementia. The P301L missense mutation was introduced into the PIMAX plasmid and confirmed with sequencing. (**C**) WT tau, WT p-tau, P301L tau, and P301L p-tau purified from *E. coli* and visualized with Coomassie blue staining. (**D**) Mass spectrometry analysis of 1N4R tau and 1N4R P301L tau. Dots represent phosphorylated residues. Note that S262 is found to be phosphorylated only in the P301L mutant, whereas several other sites (S46, T50, T111, Y197, S199, T205, T212, S320, S324, S352, T361, T386, S409, and S422) are specific to the wildtype species. Vertical bars on the top represent the S/T/Y phosphorylation sites in the tau sequence.

**Figure 2 ijms-24-14996-f002:**
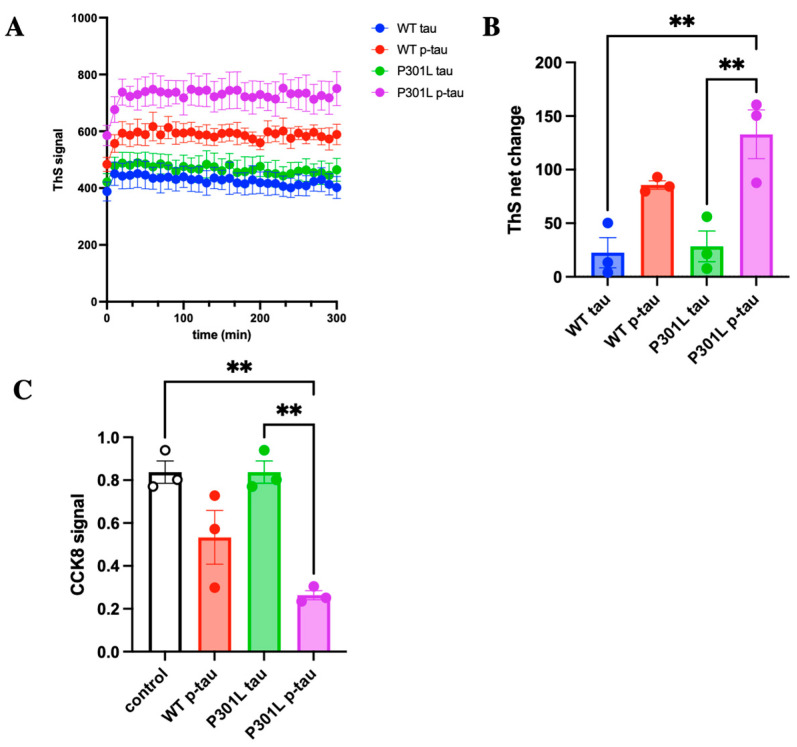
ThS aggregation assay and SH-SY5Y cell cytotoxicity assay of different tau proteins. (**A**) Kinetics curves from ThS fluorescence over 5 h. P301L p-tau presents overall the highest ThS signal. (**B**) Net change of ThS fluorescence was calculated for each sample. P301L p-tau showed the most increased ThS signal. (**C**) The viability of SH-SY5Y cells treated with 2 μM of tau proteins for 20 h was measured by the CCK8 assay. Quantification with CCK8 shows P301L p-tau is more toxic to cells than WT p-tau and unphosphorylated P301L tau. Error bars obtained from standard error. One-way ANOVA. ** *p* ≤ 0.01.

**Figure 3 ijms-24-14996-f003:**
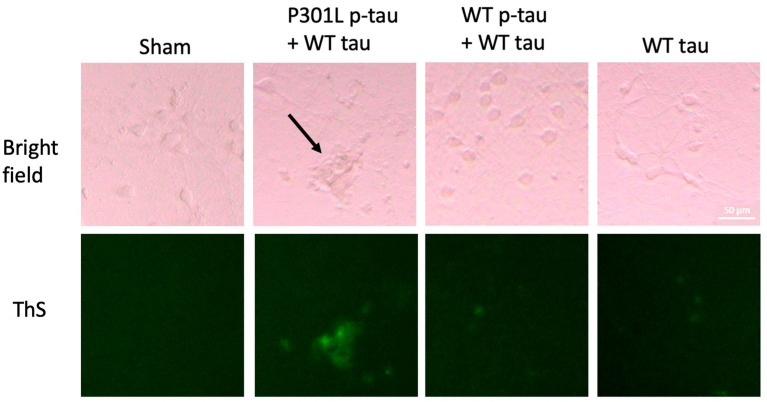
P301L p-tau shows strong seeding activity in primary neurons. Mouse primary neurons were treated with a mixture of 0.1 µM p-tau and 0.9 µM tau or 1 µM WT tau alone for three days and then stained by ThS for amyloid structures. P301L p-tau induced the strongest ThS signal and also abnormal cell morphology (black arrow), while WT p-tau had minor effects on tau aggregation and cell shape. Representative images from three technical repeats were shown. Magnification is 20×. Scale bar = 50 µm.

**Figure 4 ijms-24-14996-f004:**
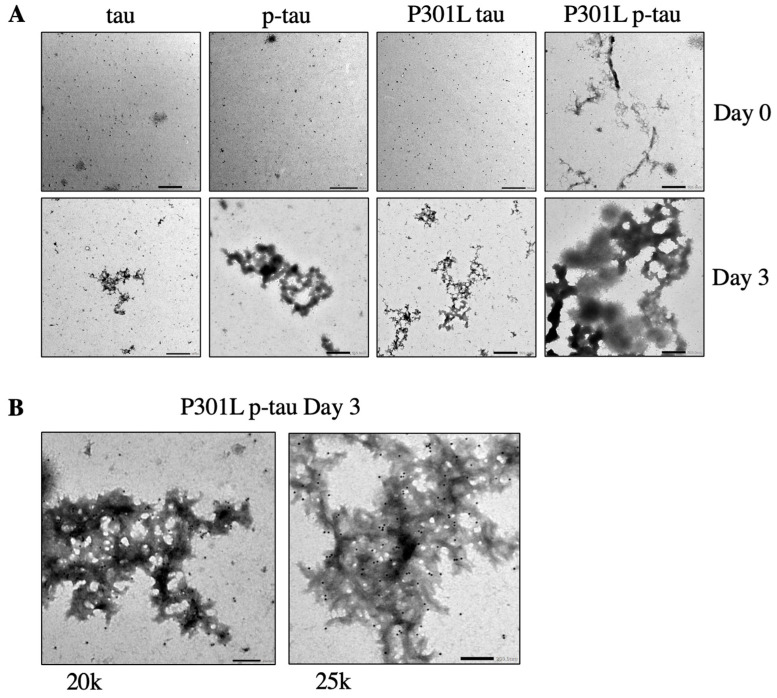
Transmission electron microscopy images of WT p-tau and P301L p-tau aggregation development. (**A**) PIMAX purified tau, p-tau, P301L tau, P301L p-tau, incubated for 0 days vs. 3 days at 10k magnification. P301L p-tau showed a more organized structure at day 0. Scale bar = 500 nm. (**B**) Anti-tau immunogold staining of PIMAX-purified P301L p-tau, incubated for 3 days, verified the structure is composed of tau. Magnification is 20k and 25k. Scale bar = 200 nm.

**Figure 5 ijms-24-14996-f005:**
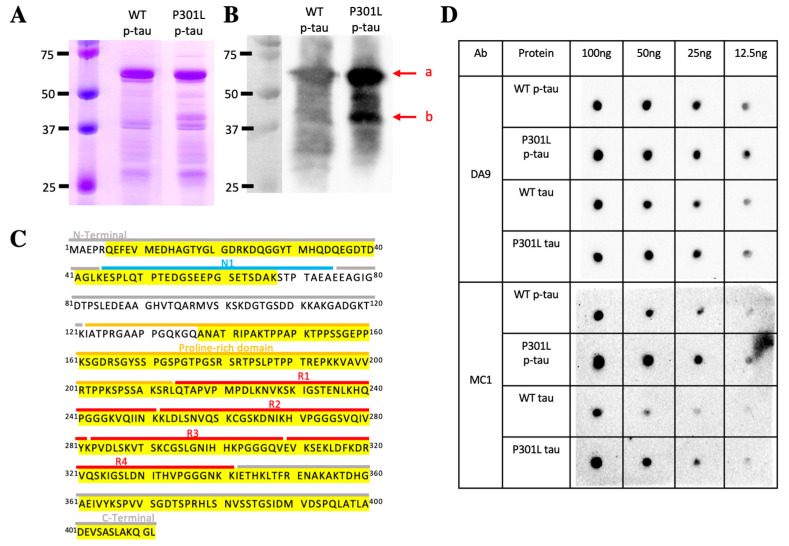
P301L p-tau adopts a specific conformation sensitive to bacterial endopeptidases. (**A**) SDS-PAGE and Coomassie blue staining images of WT p-tau and P301L p-tau. (**B**) Western blots of WT p-tau and P301L p-tau confirmed the major tau species at around 60kD. Band a: major p-tau species. Band b: truncated tau species specific to P301L p-tau. The DA9 (total tau) antibody was used as the primary antibody. (**C**) Mass spectrometry analysis showed that the band b in figure A is composed of a part of the N-terminus and C-terminus of the tau sequence (yellow highlighted). (**D**) Western dot-blot of different tau species using DA9 and MC1 antibodies. DA9 shows the total tau amount in each tau species blotted on the membrane. MC1 recognizes the specific conformation of tau that P301L p-tau presented the most epitope, followed by WT p-tau, P301L tau, and WT tau the least. Proteins were not denatured as required for SDS-PAGE to preserve the native conformation through dot-blot immunodetection.

**Figure 6 ijms-24-14996-f006:**
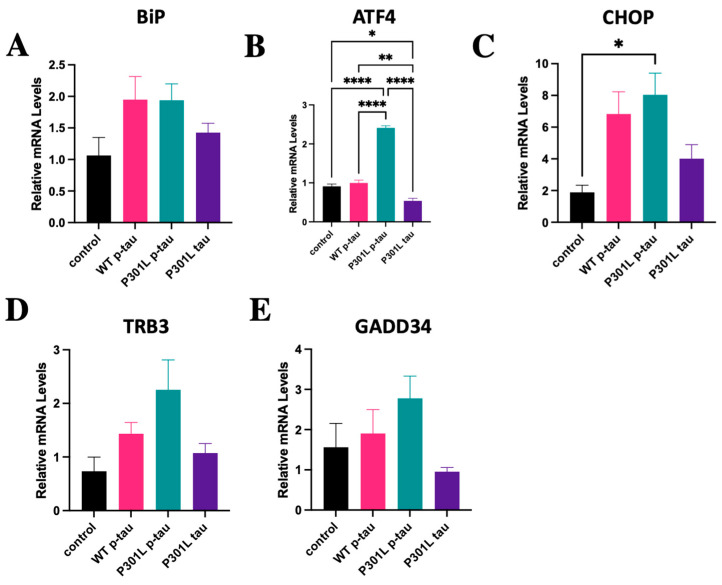
qPCR analysis for ER stress and ER stress-associated pro-apoptosis markers (BiP, ATF4, CHOP, TRB3, and GADD34) in SH-SY5Y cells treated with control (vehicle), 0.5 µM WT p-tau, P301L p-tau, or P301L tau for 24 h. P301L p-tau induced more ER stress-associated pro-apoptosis RNA levels than WT p-tau or P301L tau. Mean ± SEM (*n* = 3 repeats). One-way ANOVA. * *p* ≤ 0.05, ** *p* ≤ 0.01, **** *p* ≤ 0.0001.

**Figure 7 ijms-24-14996-f007:**
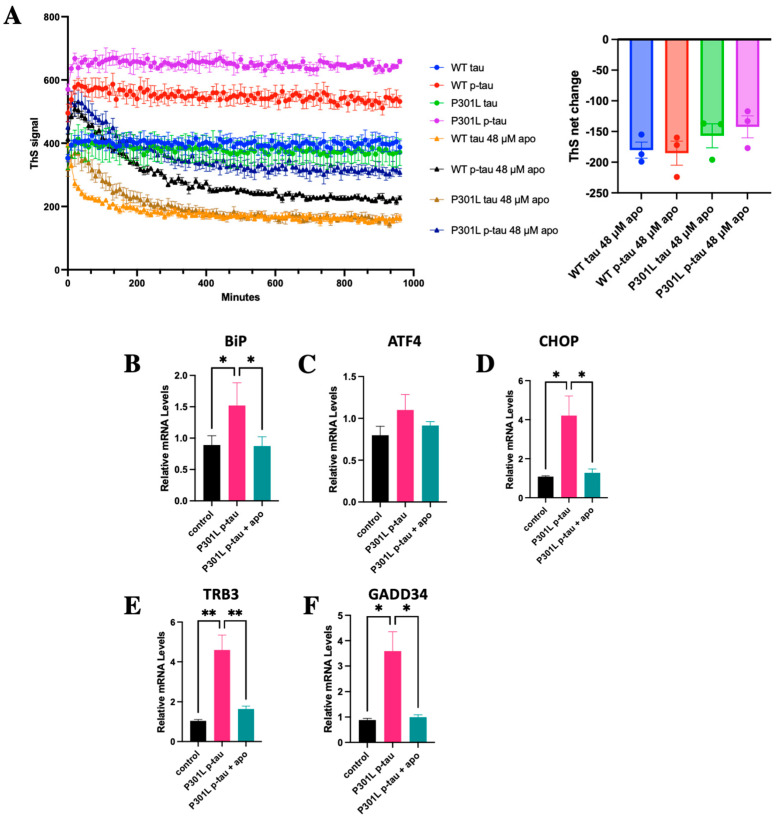
Aggregation and cytotoxicity of P301L p-tau can be ameliorated by apomorphine. (**A**) ThS aggregation assay of 6 µM p-tau or P301L p-tau in the absence or presence of 48 μM apomorphine, a p-tau aggregation inhibitor. P301L p-tau shows similar kinetic and net change responses to WT p-tau. (**B**–**F**) qPCR analysis of ER stress-associated pro-apoptosis in SH-SY5Y cells treated with control (vehicle) and P301L p-tau (0.5 µM) in the absence or presence of apomorphine (5 µM) for 24h. ER stress-associated pro-apoptosis was reduced in the presence of apomorphine. Mean ± SEM (*n* = 3 repeats). One-way ANOVA. * *p* ≤ 0.05, ** *p* ≤ 0.01.

**Figure 8 ijms-24-14996-f008:**
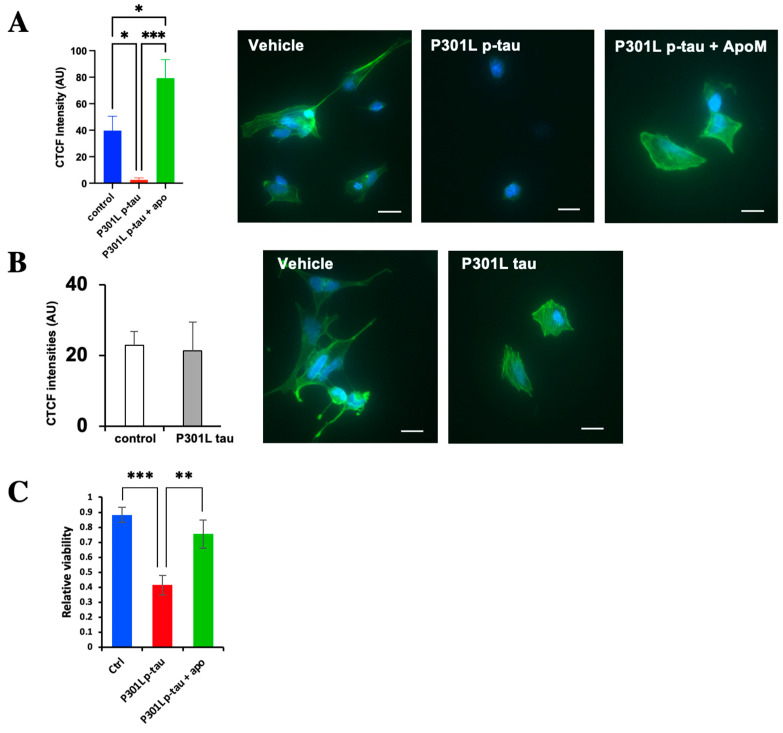
Hyperphosphorylated P301L p-tau, but not P301L tau, can disrupt the cellular cytoskeleton. (**A**) Cytoskeleton staining of SH-SY5Y treated with P301L p-tau (0.5 µM) or vehicle in the absence or presence of apomorphine (5 µM) for 24 h. (**B**) Cytoskeleton staining of SH-SY5Y cells treated with P301L tau (0.5 µM) or vehicle for 24 h. Magnification: 60× Scale bar: 20 µm “Intracellular cytoskeleton abundance, reflected by corrected total cell fluorescence (CTCF), was quantified. (**C**) Relative cell viability of SH-SY5Y cells treated with 0.5 µM P301L p-tau in the presence or absence of 5 µM apomorphine was measured by FDA/PI staining. Mean ± SEM (*n* = 4 repeats). One-way ANOVA. * *p* ≤ 0.05, ** *p* ≤ 0.01, *** *p* ≤ 0.001.

## Data Availability

All original data are available upon request.

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
