# Peer review of "Frontotemporal Dementia P301L Mutation Potentiates but Is Not Sufficient to Cause the Formation of Cytotoxic Fibrils of Tau"

_ijms, 2023, doi:10.3390/ijms241914996_

Round 1

Author Response

Dear reviewer, 

All the concerns should be addressed in the revision manuscript. Please see below for the major changes that we made to address your suggestions.

  1. If seeding experiments in SHSY5Y cells with the different tau species are able to lead to the formation of tau aggregates?

We include a new figure of seeding activity experiment with mouse primary neurons. (Page 4 and Figure 3)

  1. If they performed three independent replicates for each experiment in Figure 4?

Yes, we performed several replicates and it’s described in the manuscript. (Page 6)

  1. The authors should mention the gene used as a loading control for the quantification.

Primer list and loading control are added to the Method section. (Page 19)

  1. Quantification should be included in the figures to show the statistical significance.

All the quantification errors are fixed.

  1. The authors should include in Figure 7 a set of experiments of SH-SY5Y viability with the different proteins with and without apomorphine.

The figure is updated. The CTCF quantification shows that apomorphine protected the cells and maintained cytoskeleton integrity.

  1. Concerns regarding Coomassie blue staining and statistical analysis:

Coomassie blue staining and Statistical methods are added to the Method section.

Reviewer 2 Report

GENERAL MAJOR FEEDBACK:

The manuscript titled “Frontotemporal dementia P301L mutation potentiates but is not sufficient to cause the formation of cytotoxic fibrils of tau” by Wang et al examined the effects of the P301L mutation in the tau protein with and without hyperphosphorylation on the propensity of tau to for amyloidogenic aggregates, their cytotoxicity, and the potentially therapeutic effects of treatment with apomorphine, a known hyperphosphorylated tau aggregation inhibitor. This study is comprehensive, exciting, and impactful. While is well organized and reasonably well written, there are some opportunities for improvement, including some missing controls, and several small typographical errors, as enumerated below.

  1. Figure 2B is missing comparison lines (significance assessment) for tau vs P201L p-tau, and tau vs p-tau. Usually, when the differences are not significant, they are labeled “n/s”. Without the labels, the reader has to assume that comparisons that are not noted are not statistically significant, but there is no way to know for sure. For clarity, please add an “n/s” label for changes that are not significant, or at least describe this in the figure legend. This also applies to panel c, as well as to figures 5, 6 and 7 (basically, for all figures containing panels showing differences between bar plots).
  2. Similarly, for Figure 2B, change the label ‘tau’ to ‘WT tau’ for further clarity, and also replace the “Control” label in panel C to be more explicit (this apply also to figures 5, 6, & 7; please change the control labels to explicitly state exactly what control is being plotted, e.g., WT tau).
  3. It would be great if the authors could briefly explain in the Introduction and/or in page 5, line 201 why the specific cell line used (neuroblastoma SH-SY5Y cells) was chosen for experiments assessing cytotoxicity and cell death (as opposed to other potential cell lines).
  4. Labels are missing for the lanes in the gels in Figure 4, panel A.
  5. In Figure 4 panel D, important controls are missing. Why were WT tau and P301L (without hyperphosphorylation) not assayed for MC1 and DA9 binding? Is it the mutation or hyperphosphorylation that likely induces conformational changes, or is it both? To better substantiate the claim in the brief discussion in page 8, line 283 “It is therefore likely that certain population of hyperphosphorylated P301L tau molecules adopted a specific conformation that interfered with the formation of the MC1 epitope”, testing for antibody binding in the absence of hyperphosphorylation (for both WT tau and P301L) would help a lot.
  6. In Figure 5, panel B, are the ** and *** marks reversed? Otherwise, how is it that a the larger difference between Ctl vs P301L p-tau is less significant than the comparatively smaller change between p-tau vs P301L p-tau?
  7. In Figure 6, why are there no error bars for the p-tau and P301L p-tau data points in the plot in the absence of apo treatment? Also, why are the ThS signals much lower than in Figure 2 panel A? Also, why are the controls WT tau + apo and P301L + apo missing from this plot?

SPECIFIC NINOR FEEDBACK:

8.     In the Abstract, page 1, line 19, “defective” should be replaced with “deleterious” (or “harmful”).

9.     The word “that” in the Abstract, page 1, line 23 should be changed to “by which” (or “whereby”).

10.  “Relevancy” in page 3, line 129 should be “relevance”.

11.  “Calculate” in page 3, line 144 should be “calculated”.

12.  Page 4, line 156 briefly considers the possibility of structural differences between different tau species. It would be great if the authors could also mention in this context (and/or in the Introduction and Discussion) whether there are any existing high-resolution structures of tau, tau domains, or tau filaments or aggregates, resolved by X-ray crystallography, nuclear magnetic resonance, and/or cryogenic electron microscopy and tomography.

13.  In page 5, after line 199, it would be useful to explicitly say in writing that the increase in ThS fluorescence was statistically significant for P301L p-tau compared to P201L tau without phosphorylation.

14.  In page 8, line 272, the word “posited” should be “posit” (“…one could further extrapolate and posit that…”).

15.  In page 8, line 281, by the word “member” the authors likely meant “membrane”.

16.  In page 10 line 354, there’s no need for the hyphen in “ap-omorphine”.

17.  In page 11, line 378, “were quantified” should be “was quantified”.

18.  The word “character” (or characters) seems to be misused in several places, such as page 5 line 187, page 12 lines 409, 420, 422, and 447. These instances should likely be replaced with “characteristic” or “characteristics”.

19.  In page 12, line 441, in the part that says “and that Pro301”, the word “that” should be removed.

20.  Page 13, line 458, “in additional” should be changed to “in addition”.

21.  Please make sure there is a space between all quantities followed by units. For example “10mM” in page 14, line 506, should be “10 mM”. Other examples of this are in lines 512 and 521.

Only minor typos and problematic sentences as well as misuse of a couple of words need to be improved.

Author Response

Dear reviewer, 

All the concerns should be addressed in the revision manuscript. Please see below for the major changes that we made to address your suggestions.

  1. Concern regarding statistical significance in some of the figures:

Statistical significance problems are fixed in the figures and clarified in Method section. (Page 19)

  1. Concerns regarding labeling of tau and controls:

Misleading terms of control or tau are fixed in the figures and figure legends.

  1. Important controls are missing for Figure 4.

We apologize for the mislabeling of figures. We strived to repeat the experiments with two extra, independent batches of samples. We now provide an updated Figure 5 and related description/discussion. Please see Figure 5D (Results, lines 242 – 255), and Discussion (lines  366 – 369)

  1. Why are the controls WT tau + apo and P301L + apo missing in Figure 6?

The figure is updated with the controls. (Figure 7)

  1. Why are the ThS signals in Figure 6 much lower than in Figure 2?

The absolute values of the ThS fluorescence can vary depending on the batch of the proteins, the sample age, and even day-to-day related to the instrument. We therefore do not compare the absolute fluorescence values across experiments, but rely strictly on the relative changes compared to the control installed in each and every set of experiments. We therefore refrain from making a direction comparison between the numeric values between Figure 2 and Figure 6 (now Figure 7, with a new Figure 3 added to the revision).

Round 2

Reviewer 1 Report

Figure 2 is missing from the new draft.

The CTCF does not exlain if apomorphine and treatment with the different Tau strains is toxic, a cell viability assay can answer the question.

Figure 4 supplementary, is missing the figure legend.

Author Response

  1. The format might go off in the word file. We have figure 2 included.
  2. We have an updated revision with the viability data.
  3. We don't have supplementary figure in the manuscript, not sure if that refers to other figure.